# Soluble ST2 as a biomarker for predicting severe adverse events among pediatric patients with Mycoplasma pneumoniae pneumonia

Fangying Cheng☯, Tingting Li☯, Lei Zhang, Menghua Xu, Luxi Chen, Zhicheng Ye🄳*, Jin Xu*

Department of Clinical Laboratory, Children's Hospital of Fudan University, National Children's Medical Center, Shanghai, China

☯ These authors contributed equally to this work.
* yzcrxj@163.com (ZY); jinxu_125@163.com (JX)

## Abstract

### Aim

*Mycoplasma pneumoniae* (MP) is a leading cause of pneumonia in children. Early identification of patients at high risk is critical for improving outcomes. This study aimed to evaluate the association of soluble ST2 (sST2) with in-hospital adverse events in pediatric MP pneumonia (MPP).

### Methods

We retrospectively analyzed 147 children with MPP admitted to the Children's Hospital of Fudan University, Shanghai, China, between 01/04/2023 and 31/05/2024. Demographic, clinical, and laboratory data were collected, including sST2, inflammatory markers (CRP, PCT, IL-6), and blood cell counts. Severe adverse events were defined as in-hospital death, ICU admission, diagnosis of sepsis or use of extracorporeal membrane oxygenation.

### Results

Twelve patients experienced severe adverse events and had significantly higher sST2 levels. ROC analysis showed that sST2 predicted severe adverse events (AUC = 0.944, 95% CI 0.894–0.975, P < 0.001), with an optimal cut-off of 114.18 ng/mL (sensitivity 91.7%, specificity 94.8%). The association remained significant after adjusting for age, sex, PCT, and IL-6. In addition, admission sST2 levels were significantly higher in severe MPP cases, those with co-infections and those with pulmonary complications and/or extrapulmonary complications during hospitalization. sST2 correlated positively with hospital length of stay and preadmission fever duration. They also correlated positively with neutrophil counts, neutrophil to lymphocyte ratio, PCT, CRP and IL-6 but negatively with lymphocyte counts. **Conclusions.**

**Data availability statement:** All relevant data are within the manuscript and its Supporting information files.

**Funding:** The author(s) received no specific funding for this work.

**Competing interests:** The authors have declared that no competing interests exist.

sST2 was associated with in-hospital adverse events and it showed better performance in predicting severe adverse events than other inflammatory biomarkers. The potential of sST2 as a prognostic biomarker for MPP warrants further investigation.

## Introduction

*Mycoplasma pneumoniae* (MP) is a major cause of community-acquired pneumonia (CAP) among children in China, especially those over 5 years old [1]. Mycoplasma pneumoniae pneumonia (MPP) typically presents with sore throat, coryza, intermittent irritating cough, often accompanied by headache, fever, and myalgia. The infection occurs endemically with an epidemic peak every few years. After the COVID-19 pandemic, China experienced a notable surge of MP infections from 2023 to 2024. This epidemic was associated with more severe clinical manifestations and a high prevalence of macrolide-resistant M. pneumoniae (MRMP), which complicated treatment [2,3]. These trends highlight the importance of early identification of children at risk of severe or rapidly progressing disease to enable timely intervention.

Currently, physicians often rely on clinical judgement based on overall patient assessment to distinguish high-risk patients. Previous studies have reported mixed results regarding the accuracy of this approach as well as substantial inter-physician variability [4,5].Thus, clinical decision aids incorporating objective parameters have been proposed. However, existing tools show only moderate accuracy in pediatric respiratory infections and require further validation [6–8]. Blood biomarkers may represent a potential alternative for assessing severity and predicting prognosis.

Soluble ST2 (sST2) is a circulating form of the ST2 receptor. It competes with membrane-bound ST2 for binding to interleukin-33 (IL-33). The IL-33/ST2 pathway modulates immune response and inflammation [9]. Immune dysregulation, including T-cell activation and cytokine-driven inflammation, is believed to contribute to MPP progression [10,11]. Elevated sST2 may reflect the degree of immune activation and inflammation in MPP. Consequently, elevated sST2 may be associated with disease severity and, in turn, adverse events. Some studies in adults have linked sST2 to prognosis in community-acquired pneumonia [12,13]. Few studies have evaluated sST2 in pediatric MPP. The relationships between sST2 levels and clinical outcomes in MPP remain unclear.

Inflammatory biomarkers such as procalcitonin (PCT), C-reactive protein (CRP), and interleukin-6 (IL-6), as well as immune cell counts such as white blood cell counts (WBC), neutrophil counts (NEU), and lymphocyte counts (LYM) are widely used laboratory tests for pneumonia patients. Studies have shown that CRP, IL-6 and neutrophil-to-lymphocyte ratio (NLR) are associated with severe MPP or refractory MPP (RMPP) in children [14–17]. How sST2 compares with conventional inflammatory markers in children with MPP remains unknown. Therefore, in this study, we aimed to evaluate the association of sST2 with in-hospital adverse events and clinical characteristics in pediatric MPP patients, and to assess its potential as a prognostic biomarker in comparison with conventional inflammatory markers.

## Materials and methods

### Subjects

This study retrospectively analyzed 147 children diagnosed with Mycoplasma pneumoniae pneumonia (MPP) who were admitted to the Children's Hospital of Fudan University in Shanghai, China, between April 2023 and May 2024. The retrospective data analysis was performed from June 2024 to January 2025. The inclusion criteria were: (1) MPP diagnosis, (2) age between 1 month and 18 years, (3) hospital stay > 24 hours, and (4) available sST2 measurements at admission. The exclusion criteria were patients with (1) congenital heart diseases, (2) autoimmune diseases, (3) cancer, (4) active asthma or pulmonary tuberculosis, and (5) incomplete medical records, because sST2 levels may be elevated in these conditions, while patients with cancer and pulmonary tuberculosis were excluded because of their significant impact on prognosis.

Patients were diagnosed with MPP according to the Chinese guideline for the diagnosis and treatment of childhood *Mycoplasma pneumoniae* pneumonia (2023). The diagnostic criteria were as follows [1,2]: (1) evidence of acute respiratory tract infections, accompanied by chest radiography-proven pneumonia; (2) a fourfold or greater increase in MP antibody titers of paired sera, or > 1:160 titer of single serum MP antibody, or MP-DNA(+) or MP-RNA(+).

The sST2 assay was a newly introduced test during the study period and was performed at the attending physician's discretion. During this period, 3,794 patients with MPP were hospitalized, of whom 205 had sST2 results available. Of these, 58 patients were excluded due to the following reasons: age or hospital stay length (n = 4), sST2 assay performed after 24 hours after admission (n = 19), congenital heart diseases (n = 6), autoimmune diseases (n = 4), cancer (n = 3), and ative bronchial asthma or pulmonary tuberculosis (n = 22). Finally, 147 patients were included. Complete medical records were available for included patients (S1 Fig).

### Clinical data collection

Data were obtained from medical records, including demographic characteristics, identified pathogens, physician-diagnosed conditions, ICU admissions, and in-hospital deaths. Severe adverse events and pulmonary and extrapulmonary complications were classified according to predefined criteria. Severe MPP was defined as MPP with any of the following criteria according to the Chinese guideline for the diagnosis and treatment of childhood *Mycoplasma pneumoniae* pneumonia (2023) [1,2]: (1) high fever (> 39°C) for more than 5 days or fever for more than 7 days without a declining trend in peak temperature; (2) hypoxemia (maintained an $SaO_2 < 92\%$ on room air); (3) increasing respiratory and pulse rates with clinical evidence of respiratory distress and exhaustion with or without a elevated $PaCO_2$; (4) signs of intrapulmonary infection, such as moderate to large pleural effusion, large area of pulmonary consolidation, plastic bronchitis, pulmonary embolism, necrotizing pneumonia, and acute asthma exacerbations; and (5) signs of extrapulmonary complications, such as meningoencephalitis, ascending (i.e., Guillain-Barré) paralysis, myopericarditis, erythema multiforme, autoimmune hemolytic anemia, hemophagocytic syndrome, or disseminated intravascular coagulation. Severe adverse events were defined as in-hospital death, ICU admission, diagnosis of sepsis or use of extracorporeal membrane oxygenation (ECMO). Pulmonary complications included acute respiratory distress syndrome, respiratory failure, necrotizing pneumonia, pleural effusion, plastic bronchitis, pulmonary embolism and pulmonary atelectasis. Extrapulmonary complications included complications of nervous system, circulatory system, blood, skin and mucous, liver, kidney, muscles, and pancreas (S1 File).

### Laboratory tests

Laboratory test results including sST2, CRP, PCT, IL-6, WBC, LYM, NEU and NLR at admission were collected from medical records. All tests were performed using venous blood samples collected at admission. sST2 and IL-6 were measured by fluorescent immunoassays on Pylon 3D immune analyzer from ET Healthcare Inc. (Suzhou, China) using EDTA-anticoagulated plasma. PCT was measured by electrochemiluminescence assays using the instruments and

reagents from Roche Diagnostics GmbH (Mannheim, Germany) with serum. CRP was measured by turbidimetric immunoassays using the instruments and reagents from Shanghai Upper Bio-Tech Pharma Ltd. (Shanghai, China) with EDTA-anticoagulated plasma. Complete blood counts were acquired by an XN-2000 analyzer using reagents from Sysmex Corp.(Kobe, Japan) with EDTA-anticoagulated whole blood. All the assay kits and analyzers were approved by the China National Medical Products Administration (NMPA) as in-vitro diagnostic products.

### Study variables

The outcome variable of this study was severe adverse events, defined as in-hospital death, ICU admission, diagnosis of sepsis or use of ECMO (S1 File). The primary predictor variable was sST2, and the covariates included demographic variables (age, sex) as well as other blood biomarkers (CRP, PCT, IL-6, WBC, LYM, NEU and NLR).

### Statistical analysis

Continuous variables were presented as median (interquartile range) and compared by Mann-Whitney U test or Kruskal-Wallis test. A post hoc Conover test was conducted after a significant Kruskal-Wallis test. Categorical variables were expressed as numbers (%) and compared using Fisher's exact test. Spearman correlation was used for correlation analyses between sST2 and other parameters. The optimal cutoff for sST2 was determined by maximizing Youden's index from receiver operating characteristic (ROC) curve analysis. The areas under the ROC curve (AUC) were calculated to compare the predictive performance of biomarkers for severe adverse events. Multivariable logistic regression assessed the independent prognostic value of sST2 after adjustment for age and sex or other biomarkers. All statistical tests were two-tailed, and a value of $P < 0.05$ was considered statistically significant. Statistical analyses were performed using MedCalc Statistical Software version 22.009 (MedCalc Ltd., Ostend, Belgium). All analyses were performed on complete observations given that there were no missing values present in the data.

### Ethics statement

This study was conducted in accordance with the Declaration of Helsinki and was approved by the Ethics Committee of the Children's Hospital of Fudan University (Ethics No. 2024−129). We confirm that all methods were performed in accordance with the relevant guidelines and regulations. This study is a retrospective analysis, only involving the statistical analysis of patient data, and all patient information has been anonymized and does not contain sensitive data; therefore, the requirement for informed consent was waived by the ethics committee.

## Results

### Patient characteristics

147 children (77 females, 52.4%) hospitalized with MPP were included, of whom the median age was 6 years, and median hospital stay was 7 days. Severe adverse events occurred in 12 patients (8.2%), including 1 death, 10 ICU admissions, 9 patients diagnosed with sepsis and 4 patients requiring ECMO during their hospital stay. The demographic data, clinical parameters and laboratory results at admission of all the study subjects were presented in Table 1 and compared between patients with and without severe adverse events. The two groups had similar sex distribution and pre-admission fever duration. However, in the group with severe adverse events, patients were significantly younger and had lower body weight. There were also significantly higher proportions of severe MPP patients and patients with pulmonary and extrapulmonary complications in the group of severe adverse events, although the proportions of RMPP/MRMP and co-infection with other pathogens were similar between the two groups. The group with severe adverse events had a significantly longer hospital stay.

**Table 1. Patient characteristics.**

| Characteristics[1] | Overall (N = 147) | Severe Adverse Events | | P-value[2] |
|---|---|---|---|---|
| | | No (N = 135) | Yes (N = 12) | |
| Sex | | | | 0.135 |
| Female | 77 (52.4%) | 68 (50.4%) | 9 (75.0%) | |
| Male | 70 (47.6%) | 67 (49.6%) | 3 (25.0%) | |
| Age (years) | 6.0 (4.0, 9.0) | 6.0 (4.0, 9.0) | 4.0 (1.8, 6.0) | 0.047 |
| Weight (kg) | 23.0 (16.5, 31.0) | 23.2 (17.0, 31.0) | 15.50 (12.8, 20.0) | 0.018 |
| Severe MP | 46 (31.3%) | 34 (25.2%) | 12 (100.0%) | <0.001 |
| RMPP/MRMP | 51 (34.7%) | 47 (34.8%) | 4 (33.3%) | 0.999 |
| Co-infections | 83 (56.5%) | 74 (54.8%) | 9 (75.0%) | 0.231 |
| Bacterial infection | 21 (14.3%) | 18 (13.3%) | 3 (25.0%) | 0.38 |
| Viral infection | 72 (49.0%) | 63 (46. 7%) | 9 (75.0%) | 0.074 |
| Fungal infection | 5 (3.4%) | 3 (2.2%) | 2 (16. 7%) | 0.053 |
| Complications | | | | <0.001 |
| None | 99 (67.4%) | 99 (73.3%) | 0 (0.0%) | |
| Extrapulmonary | 14 (9.5%) | 14 (10.4%) | 0 (0.0%) | |
| Pulmonary | 24 (16.3%) | 19 (14.1%) | 5 (41. 7%) | |
| Pulmonary and extrapulmonary | 10 (6.8%) | 3 (2.2%) | 7 (58.3%) | |
| Pulmonary complications | 34 (23.1%) | 22 (16.3%) | 12 (100.0%) | <0.001 |
| ARDS | 2 (1.4%) | 0 (0.0%) | 2 (16. 7%) | 0.006 |
| Respiratory failure | 11 (7.5%) | 2 (1.5%) | 9 (75.0%) | <0.001 |
| Pulmonary necrosis | 3 (2.04%) | 1 (0.7%) | 2 (16.7%) | 0.018 |
| Pleural effusion | 11 (7.5%) | 9 (6.7%) | 2 (16.7%) | 0.222 |
| Plastic bronchitis | 2 (1.4%) | 2 (1.5%) | 0 (0.0%) | 0.999 |
| Pulmonary embolism | 1 (0.7%) | 1 (0.7%) | 0 (0.0%) | 0.999 |
| Pulmonary consolidation | 54 (36.7%) | 52 (38.5%) | 2 (16. 7%) | 0.211 |
| Pulmonary atelectasis | 11 (7.5%) | 10 (7.4%) | 1 (8.3%) | 0.999 |
| Extrapulmonary complications | 24 (16.33%) | 17 (12.59%) | 7 (58.33%) | <0.001 |
| Nervous system | 4 (2.7%) | 1 (0.7%) | 3 (25.0%) | 0.002 |
| Circulatory system | 10 (6.8%) | 7 (5.2%) | 3 (25.0%) | 0.036 |
| Blood system | 5 (3.4%) | 3 (2.2%) | 2 (16.7%) | 0.053 |
| Skin and mucous | 5 (3.4%) | 5 (3.7%) | 0 (0.0%) | 0.999 |
| Other systems[3] | 5 (3.4%) | 2 (1.5%) | 3 (25.0%) | 0.004 |
| Fever duration before hospital admission (days) | 7.0 (5.0, 9.0) | 7.0 (5.0, 9.0) | 8.0 (5.0, 10.5) | 0.415 |
| Hospital stay (days) | 7.0 (5.0, 9.0) | 6.0 (5.0, 9.0) | 17.5 (11.0, 37.0) | <0.001 |
| ST2 (ng/mL) | 39.3 (16.5, 84.6) | 34.0 (15.7, 68.0) | 170.7 (132.5, 246.3) | <0.001 |
| WBC (10^9/L) | 7.5 (5.6, 10. 6) | 7.5 (5.6, 10.4) | 9.0 (5.3, 15.1) | 0.408 |
| NEU (10^9/L) | 4.6 (3.1, 6.6) | 4.46 (3.1, 6.4) | 6.92 (3.8, 12.9) | 0.073 |
| LYM (10^9/L) | 2.0 (1.5, 2.8) | 2.1 (1.6, 3.2) | 1.5 (1.2, 1.8) | 0.01 |
| NLR | 2.1 (1.2, 3.7) | 2.0 (1.2, 3.4) | 5.7 (2.1, 8.1) | 0.004 |
| CRP (mg/L) | 10.9 (2.3, 24.0) | 7.4 (2.0, 22.0) | 48.8 (18.5, 58.6) | <0.001 |
| PCT (ng/mL) | 0.09 (0.04, 0.20) | 0.08 (0.04, 0.17) | 0.85 (0.23, 1.80) | <0.001 |
| IL-6 (pg/mL) | 11.0 (4.2, 27.4) | 9.7 (3.6, 21.6) | 40.1 (28.3, 65.1) | <0.001 |

1 n (%); Median (Q1, Q3) [2]Fisher's exact test; Mann-Whitney U test [3]Complications in liver, kidney, muscles, or pancreas.

## Correlation of sST2 with other laboratory biomarkers and clinical parameters

sST2 levels were correlated positively with NEU (rho = 0.165, P = 0.045), NLR (rho = 0.384, P < 0.001), PCT (rho = 0.386, P < 0.001), CRP (rho = 0.378, P < 0.001) and IL-6 (rho = 0.211, P = 0.010) and negatively correlated with LYM (rho = −0.317, P < 0.001) (S2 Fig). We also found sST2 at admission was positively correlated with hospital length of stay (rho = 0.632, P < 0.001) and pre-admission fever duration (rho = 0.198, P = 0.017) (S3 Fig).

Patients with severe MPP and MPP patients with co-infection by other respiratory tract pathogens had significantly higher sST2 levels (severe MPP vs non-severe MPP: 71.91 [30.76–140.12] vs 26.2 [15.15–54.69], P < 0.001; MPP patient with co-infection vs those without: 48.46 [23.23–94.17] vs 23.47 [13.18–52.87] ng/mL, P < 0.001) (Fig 1). However, sST2 levels were similar between patients with or without RMPP/MRMP (45.97 [17.00–87.01] vs 34.39 [13.90–79.26] ng/mL, P = 0.379) (S4 Fig). We also found patients with pulmonary complications and/or extrapulmonary complications during hospitalization had significantly higher sST2 levels at admission compared to those without complications (77.84 [29.56–147.16] vs 26.68 [14.53–54.54] ng/mL, P < 0.001) (Fig 2A). Pairwise comparisons showed that patients with both pulmonary and extrapulmonary complications had the highest sST2 (170.15 [140.12–245.95] ng/mL) followed by those with only pulmonary complications (58.80 [28.79–91.93] ng/mL), while sST2 levels were similar in those without complications (26.68 [14.53–54.54] ng/mL) and patients with only extrapulmonary complications (41.78 [14.32–100.05] ng/mL) (Fig 2B).

## Associations of sST2 with severe adverse events during hospital stay

Patients with severe adverse events had significantly higher sST2 levels than those without severe adverse events (170.69 [132.53, 246.26] vs 34.03 [15.69, 67.99] ng/mL, P < 0.001) (Fig 3A). Patients with severe adverse events also had higher CRP, PCT and IL-6, lower LYM and higher NLR (all P < 0.05, Table 1). Meanwhile, WBC and NEU did not differ significantly between the two groups.

ROC analysis showed that sST2 predicted patients with severe adverse events (AUC = 0.944, 95% CI 0.894–0.975, P < 0.001), with an optimal cut-off of 114.18 ng/mL (sensitivity 91.7%, specificity 94.8%) determined by Youden's index. The AUC of sST2 was higher than that of PCT (0.944 vs 0.900, P = 0.427) and that of IL-6 (0.944 vs 0.858, P = 0.101) without statistically significant differences, but the AUC of sST2 was significantly higher than those of CRP (AUC: 0.944 vs 0.822, P = 0.027), LYM (AUC: 0.944 vs 0.724, P < 0.001) and NLR (AUC: 0.944 vs 0.750, P = 0.025) (Fig 3B). Logistic regression analysis showed that increasing sST2 was associated with increased risk of severe adverse events (unadjusted OR = 1.023 per 1 ng/mL increase, 95% CI 1.012–1.033, P < 0.001). The association remained significant after

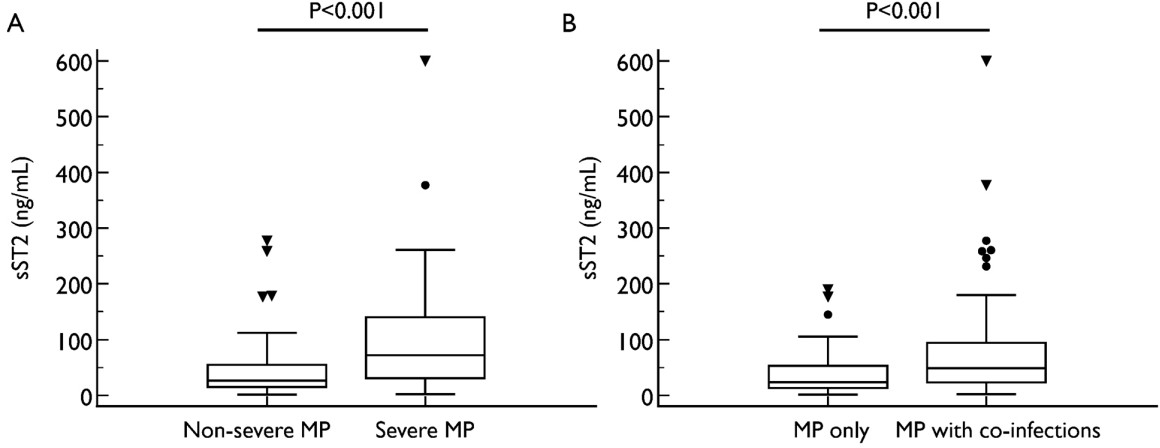

**Fig 1. Comparison of ST2 levels in patients of different severity and different pathogens.**

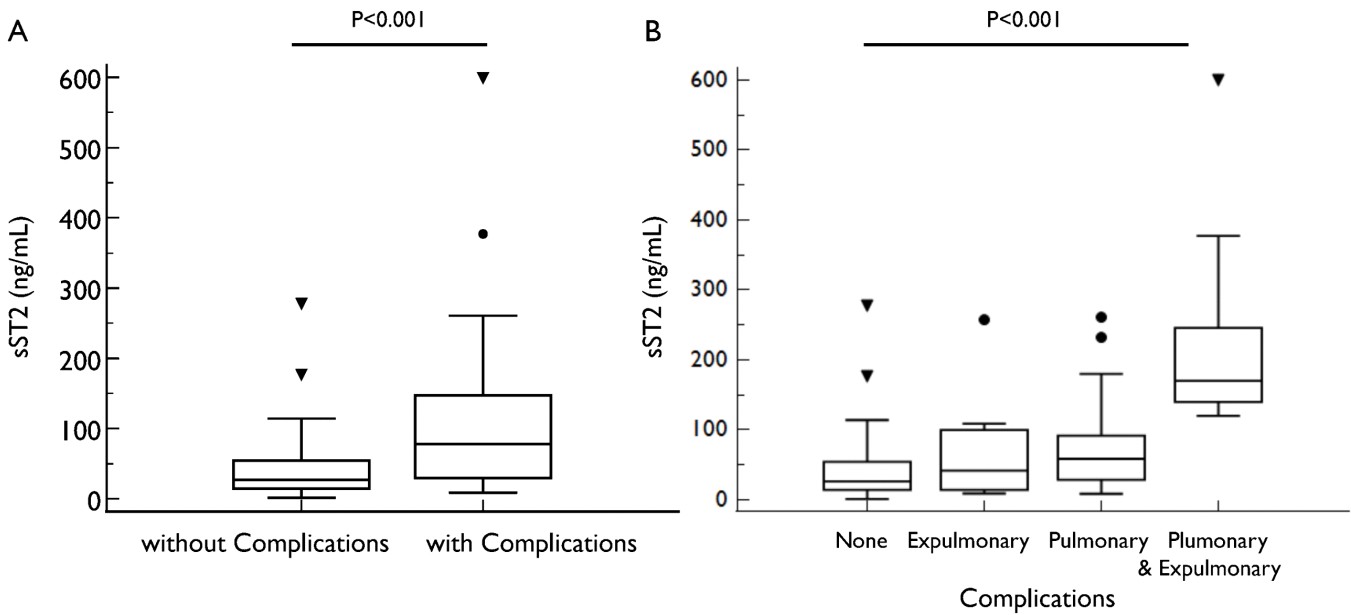

**Fig 2. Comparison of sST2 levels in patients with and without complications (A) and different types of complications (B).**

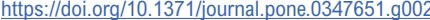
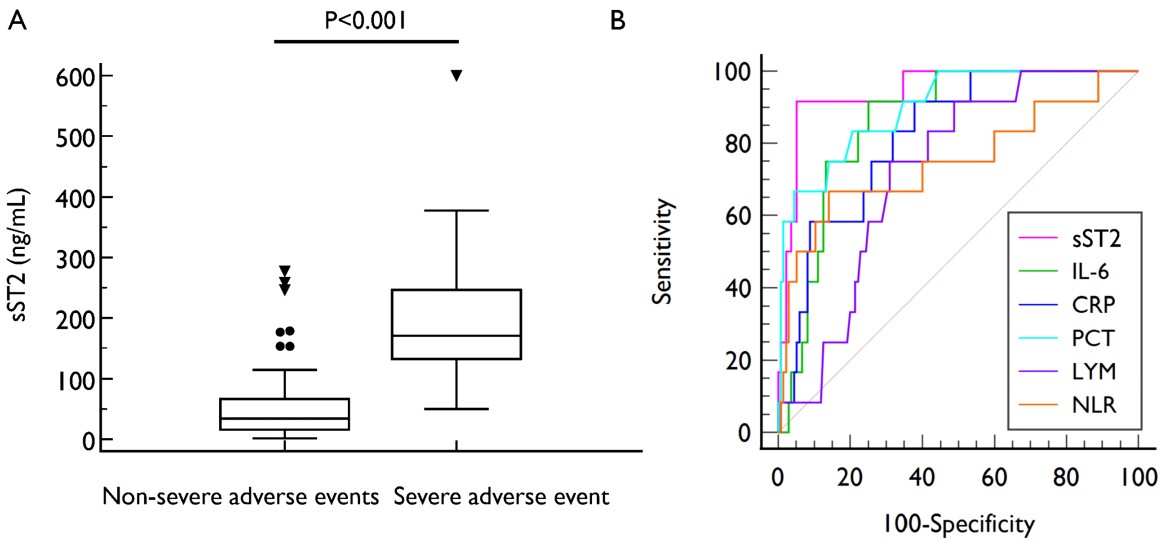

**Fig 3. Comparison of sST2 levels in patients with and without severe adverse events (A) and comparison of ROC curves of different biomarkers in differentiating patients with severe adverse events (B).**

adjusting for age and sex (adjusted OR= 1.023, 95% CI 1.022–1.034, P<0.001) or PCT and IL-6 (adjusted OR= 1.025, 95% CI 1.013–1.037, P<0.001) (Table 2).

## Discussion

The prolonged implementation of non-pharmaceutical interventions to tackle the COVID-19 pandemic between 2020 and 2022 led to a significant "immunity debt". Consequently, MP caused a prolonged non-seasonal epidemic during 2023 and

**Table 2. Multivariant logistic regression of ST2 on prediction of severe adverse events.**

|  | Odds ratio | 95% CI | P-value |
|---|---|---|---|
| Single variant model |  |  |  |
| sST2 | 1.023 | 1.012 to 1.033 | < 0.001 |
| Multivariant model 1 |  |  |  |
| sST2 | 1.023 | 1.012 to 1.034 | < 0.001 |
| Age | 0.896 | 0.697 to 1.152 | 0.371 |
| Sex = "Male" | 0.346 | 0.059 to 2.040 | 0.452 |
| Multivariant model 2 |  |  |  |
| sST2 | 1.025 | 1.013 to 1.037 | < 0.001 |
| IL-6 | 1.006 | 0.997 to 1.016 | 0.181 |
| PCT | 0.869 | 0.607 to 1.246 | 0.445 |

2024 across mainland China [18]. In this retrospective study, we evaluated sST2 as a prognostic biomarker in children diagnosed with MPP, and found that higher sST2 levels were associated with severe MPP, co-infections, and complications. More importantly, patients who experienced severe adverse events during hospitalization had significantly higher sST2 levels at admission. ROC analysis demonstrated that sST2 showed good predictive performance for these events. The prognostic value of sST2 remained significant after adjusting for age, sex, or other biomarkers. These findings support the potential of sST2 as a prognostic biomarker for risk stratification in pediatric MPP patients. Similarly, in adult CAP, sST2 has been shown to predict clinical stability, in-hospital mortality, and ICU admission, and improve the predictive performance of CURB-65 or PSI scores [12,13]. In children with severe pneumonia, sST2 at admission was associated with a higher 6-month readmission rate [19]. In children with SARS-CoV-2 related Multisystem Inflammatory Syndrome in Children (MIS-C), sST2 was associated with ICU admission [20]. It should be noted that sST2 is not a disease-specific marker, therefore it should be interpreted as a prognostic indicator rather than a diagnostic marker. To our knowledge, this is the first study to suggest sST2 as a predictor of short-term prognosis in pediatric MPP.

Previous studies have demonstrated that conventional inflammatory biomarkers are useful in risk stratification of children with pneumonia, including MPP. A previous study in pediatric CAP patients found that CRP and PCT may be useful in predicting the most severe outcomes, although they have limited use in discriminating non-severe from severe disease in children [21]. However, only 7.3% of cases in that study involved MPP. The levels of inflammatory biomarkers varied between CAP of different pathogens [22]. Studies in MPP found that lactate dehydrogenase has clinical value in predicting MP necrotizing pneumonia [23], and NLR and CRP were independent risk factors for severe MPP in children [24]. However, few studies have investigated the value of inflammatory biomarkers in predicting severe adverse events in pediatric MPP. We found that sST2 positively correlated with other inflammatory biomarkers and it showed better performance in predicting severe adverse events in this cohort based on AUC comparisons. The prognostic value of sST2 may be explained by the mechanism of sST2 involvement in modulating immune regulation and inflammation.

sST2 is a decoy receptor of IL-33, and the IL-33/ST2 axis plays a role in immune regulation [24]. IL-33 binding to membrane-bound ST2 enhances Th2 response. Notably, Th1 and Th17 cells both express sST2, and increased sST2 production promotes the Th1/Th17 response and reduces Th2/Treg response through sequestering IL-33 [25]. In addition, an in vitro study showed proinflammatory cytokines such as IL-1β and TNF-α also increase sST2 production in human lung epithelial cells [26]. High sST2 levels may reflect severe immune response and inflammation. Excessive immune activation is thought to drive severe pulmonary and extrapulmonary manifestations in MPP [18]. This may partly explain the association between sST2 and disease severity, complications, and severe adverse events. IL-33, as an alarmin, is released early during cell stress and damage, and it can activate the production of sST2 in mast cells or T cells [26]. The early production after the immune response may explain the advantage of sST2 at admission over other inflammatory

biomarkers in risk stratification. Additionally, recent data from adults with COVID-19 and children with severe pneumonia also suggest that sST2 is a dual cardio-inflammatory biomarker related to both myocardial injury and severity of inflammation and has prognostic value [20,27]. In accordance, in our cohort, those with circulatory complications (i.e., abnormal cardiac biomarkers, tachycardia, septic shock, and myocarditis) had significantly higher sST2 levels compared to the rest (107.15 [91.88–178.97] vs. 34.75 [15.74–72.21] ng/mL, P < 0.001). This may also contribute to its predictive performance on prognosis. Further, the association of sST2 and severe adverse events also indicated that the IL-33/ST2 axis could represent a potential therapeutic target for MPP. Additional studies are required to determine whether modulation of this pathway could influence disease progression.

There are limitations of this study. First, the sample size was relatively small, with only 12 severe adverse events. This increased the risk of model overfitting and precluded robust adjustments for multiple confounders simultaneously. Second, the validated clinical severity scores (e.g., pediatric early warning scores) or oxygenation indices are absent in our cohort. This prevented direct comparison of the prognostic performance of sST2 against established clinical assessment tools to determine its incremental value. Furthermore, as a single center retrospective study conducted during a unique post-pandemic epidemic, the cohort may have selection bias and the findings may reflect a unique epidemiological context, potentially limiting the generalizability. In the future, we plan to conduct a multi-center prospective study with a larger study population and comprehensive clinical data to validate the clinical value of sST2 in children with pneumonia.

## Conclusions

In this retrospective study of pediatric MPP patients, we found that sST2's association with disease severity and it showed better performance in predicting severe adverse events than other inflammatory biomarkers. sST2 is a potential a prognostic biomarker in children with MPP which warrants further investigation.

## Supporting information

**S1 File. Definitions and classifications of severe adverse events and complications.**
(PDF)

**S1 Fig. Flow diagram of study election process.**
(PDF)

**S2 Fig. Associations of ST2 with other biomarkers.**
(PDF)

**S3 Fig. Associations of ST2 with days of hospital stay and days of pre-admission fever.**
(PDF)

**S4 Fig. Comparison of ST2 levels in patients with and without RMPP/MRMP.**
(PDF)

**S1 Data. The pediatric MPP dataset in EXCEL file format.**
(XLSX)

## Author contributions

**Formal analysis:** Menghua Xu.

**Investigation:** Lei Zhang.

**Methodology:** Luxi Chen.

**Writing – original draft:** Fangying Cheng, Tingting Li.

**Writing – review & editing:** Zhicheng Ye, Jin Xu.

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
