## [Decision Letter · Decision Letter 0]

20 Jan 2026

Dear Dr. YE,

We look forward to receiving your revised manuscript.

Kind regards,

Shivkumar Gopalakrishnan, MD

Academic Editor

PLOS One

Journal Requirements:

For additional information about PLOS ONE ethical requirements for human subjects research, please refer to http://journals.plos.org/plosone/s/submission-guidelines#loc-human-subjects-research....

5. We notice that your supplementary [figures] are included in the manuscript file. Please remove them and upload them with the file type 'Supporting Information'. Please ensure that each Supporting Information file has a legend listed in the manuscript after the references list.

6. Please ensure that you refer to Figure 2 in your text as, if accepted, production will need this reference to link the reader to the figure.

7. We note you have included a table to which you do not refer in the text of your manuscript. Please ensure that you refer to Table 2 in your text; if accepted, production will need this reference to link the reader to the Table.

8. Please include a copy of Table 3, which you refer to in your text on page 15.

Additional Editor Comments :

Dear Dr ZHICHENG YE,

Your article is well conceived and holds promise. However, there are a few uncertainties which demand clarification.

1. Severe adverse events were defined as in-hospital death, ICU admission, diagnosis of sepsis or use of extracorporeal membrane oxygenation. Please specify the criteria used to define these, and what about the need for vent support, RRT, etc.

2. In some patients, sepsis might be mild without MODS, others severe sepsis with MODS. Was there any subgroup analysis performed and why if not?

3. You have mentioned that sST2 levels were significantly higher in severe MPP cases and those with co-infections. If co-infections were detected, what is the specificity for its association with MPP? Sst2 could have been elevated due to other infections as well. How do you explain the uncertainty?

4. You have mentioned that sST2 levels correlated positively with neutrophil count, neutrophil to lymphocyte ratio, PCT, CRP and IL-6- comparative analysis. Was sST2 more specific, more sensitive, cost effective, etc. What is the distinct advantage of doing Sst2, compared to performing NLR, ANC, PCT, CRP?

5. Lines 49-50: Studies reported mixing findings on accuracy of this method to predict patients’ outcome and inter-physician variations [4,5]- Grammar error needs to be corrected.

6. What is authenticity of definition of severe MPP? Your aim was to evaluate a biomarker whether it predicts severity. If only 4 parameters are assessed, there is a large gap of knowledge about those patients who were oxygen dependent, MODS but did not require organ support, those left with complications, destroyed lungs, etc.

7. Lines 83-84: The diagnosis of SMPP is made in MPP patients with either of the following criteria [2]- Grammar error needs to be corrected.

8. Line 140: media hospital stay was 7 days- grammar error

9. Line 142: ECOM- spelling error needs to be corrected.

10. The study design is ambiguous- criteria for severe MPP is delineated, however, the parameter Sst2 was evaluated only for 12 patients who had fit into 4 separate set of “adverse events”. These adverse events are overlapping over the defined criteria. It is unclear about sst2 levels among the actual population of severe MPP patients who qualified based on set criteria [2]. The study is actually assessing predictive capacity of sst2 to identify severe adverse events among MPP rather than severity of MPP.

11. Line 229: Consequently, MP infection undergone a prolonged non-seasonal epidemic, which emerged in April 2023- grammar error needs correction.

12. Title: Soluble ST2 is a potential biomarker for risk stratification of pediatric patients with mycoplasma pneumoniae pneumonia- better stated as sst2 as a biomarker for predicting severe adverse events among pediatric patients with MPP.

Thank you, with regards, Dr SHIVKUMAR GOPALAKRISHNAN, MD.,

Reviewers' comments:

Reviewer's Responses to Questions

**Comments to the Author**

1. Is the manuscript technically sound, and do the data support the conclusions?

Reviewer #1: Partly

Reviewer #2: Yes

Reviewer #3: Partly

2. Has the statistical analysis been performed appropriately and rigorously?

Reviewer #1: No

Reviewer #2: Yes

Reviewer #3: I Don't Know

3. Have the authors made all data underlying the findings in their manuscript fully available?

Reviewer #1: Yes

Reviewer #2: Yes

Reviewer #3: No

4. Is the manuscript presented in an intelligible fashion and written in standard English?

Reviewer #1: No

Reviewer #2: Yes

Reviewer #3: Yes

Reviewer #1: The study explores the association of sST2 with in-hospital adverse events and clinical characteristics in pediatric MPP patients, and assesses its potential as a prognostic biomarker in comparison with conventional inflammatory markers. The work is sound but requires major revisions before publication.

Methods

A brief description of the study setting will be appropriate. Clarify if it's routine to have sST2 as a baseline investigation for MPP in the hospital.

Sampling: Throw more light on sampling. How many confirmed MPP in the study period, and how many were excluded. Include a flow diagram (as per PLOS ONE guidelines) showing total confirmed cases of MPP recorded in the study period, how many folders were reviewed, how many excluded, and the final sample analyzed.

Clearly define and specify (primary versus secondary outcome variables, factor variables) all variables. Consider listing all variables, measurement methods, and units for clarity.

Clarify how missing data were handled.

Analysis and Results

The whole table 1 lacks clarity. Some entries do not correspond with the labelling of the columns or rows. Authors should clarify why Table 1 contains ‘severe adverse event’ in both rows and columns. Also, the decimals with excess zeroes are not necessary for frequencies in the table. Clearly specify which values represent mean (sd), median (iqr), or frequency (%), in the table. Define what constitutes ‘other infections’ in the table?

Format and standardize the table and ensure statistical reporting follows PLOS ONE’s reporting standards.

The question remains, what is/are your primary outcome measure (s) (end-point (s))? Severe MPP? Severe adverse events? Complications?

Consider fitting logistic regression for Severe MPP if it is considered a primary outcome.

Recommendation: Your introduction section should include a conceptual framework to clarify the multiple relationships under study. For instance, how presenting disease severity may be associated with high levels of sST2, how sST2 may in turn be related to in-hospital adverse events, and/or complications.

Discussion

The discussion requires focus and direction. Provide clearer comparisons with existing evidence, focusing more on the association between sST2 and the primary outcome(s).

Minor

Minor typographical and grammatical errors include wrong spellings and tenses (E.g. Line 62: Studies have shown and not ‘have showed’), missing letters; the whole manuscript requires proofreading before resubmission.

Reviewer #2: This study evaluates soluble ST2 (sST2) as a prognostic biomarker in pediatric Mycoplasma pneumoniae pneumonia and addresses a clinically important and timely question. The authors show that sST2 is strongly associated with disease severity, complications, and in-hospital adverse events, with superior predictive performance compared to several conventional inflammatory markers. The study is well conducted and biologically plausible; however, its retrospective single-center design, small number of severe adverse events, and lack of clinical severity indices limit generalizability and warrant cautious interpretation. Overall, the findings are promising and support the need for larger, prospective, multicenter studies to validate sST2 for clinical risk stratification.

Reviewer #3: Introduction

The Introduction is relevant but needs improvement

Makes Sentences more clear and simple ,

Avoid using terms like “clinical gestalt” and better to be replaced with standard language,

Prefer to use more neutral tone and avoid assertive, strong statements

The research gap is clearly defined; but could be strengthened by briefly stating what is unknown about sST2 in pediatric MPP.

Correction of minor language and formatting issues also needed

Materials and Methods

The Materials and Methods section is looks clear,

clarification of the study timeline should be done.

explain the reasons for exclusion criteria,

specify the source of diagnostic guidelines

details on the laboratory methods used should be provided especially information about sST2 measurement

make the ethics statement more concise and properly formatted.

The Clinical Data Collection

provide basic descriptions but lack essential methodological details.

provide more details about The data collection and extraction, how complications and adverse events were defined and classified.

Some outcomes like sepsis needs to be referenced to established criteria,

More clear categorization of the complications is needed. cardiac complications—such as myocarditis, pericarditis, arrhythmias, and heart failure—should be clearly highlighted, given some evidence linking soluble ST2 to cardiac injury in recent studies.

Laboratory Tests

Provide information on sample timing, assay characteristics, units, and data handling.

Both subsections require clearer definitions, standardized terminology

Correct grammatical and formatting issues

Results

The Results section contains valuable data but

Sentence needs to be more clear with correction of the language errors

Write the result with more direct and objective way

More clear and correct terminology should be used

Tables and figures need better organization, and more detailed sand clear tatistical reporting is needed

Discussion

Although the discussion is well-organized, but should be written in more clear, direct objective way aligning with the nature of this single-center, retrospective study.

Present any potential therapeutic or predictive outcomes as theoretical possibilities or areas for future investigation, not as current treatment protocols or guaranteed outcomes.

Conclusions avoid statement that indicate accuracy and superiority.

limitations should be extended to properly reflect design and analytical difficulties.

.

Reviewer #1: **Yes:** Muhyideen BashirMuhyideen BashirMuhyideen BashirMuhyideen Bashir

Reviewer #2: No

Reviewer #3: No

---

## [Author Response · Author response to Decision Letter 1]

25 Mar 2026

We have reformatted the manuscript according to the two style templates.

The ORCID iDs for the corresponding authors have been updated via submission system

（https://orcid.org/my-orcid?orcid=0009-0009-6967-6969，ORCID：0009-0009-6967-6969）.

This study was approved by the Ethics Committee of the Children’s Hospital of Fudan University (Ethics No:2024-129), and need for consent was waived by the ethics committee. We have included an Ethics Statement subsection in the Materials and methods: “This study was conducted in accordance with the Declaration of Helsinki and was approved by the Ethics Committee of the Children’s Hospital of Fudan University (Ethics No. 2024-129). We confirm that all methods were performed in accordance with the relevant guidelines and regulations. This study is a retrospective analysis, only involving the statistical analysis of patient data, and all patient information has been anonymized and does not contain sensitive data; therefore, the requirement for informed consent was waived by the ethics committee.”

The same text has been added to the “Ethics Statement” field of the submission form via the submission system.

As described above, the full ethics statement has been provided in the ‘Materials and methods-Ethics Statement’ and via the online submission system.

5. We notice that your supplementary [figures] are included in the manuscript file. Please remove them and upload them with the file type 'Supporting Information'. Please ensure that each Supporting Information file has a legend listed in the manuscript after the references list.

The supplementary figures have been removed from the manuscript file and a list of supplementary figure captions has been added in the manuscript after the reference list. A supporting document named S1-4_Fig.pdf is uploaded via the submission system.

6. Please ensure that you refer to Figure 2 in your text as, if accepted, production will need this reference to link the reader to the figure.

It was a typo and we have corrected it. Figure 2 is now referred to in ‘Results- Correlation between sST2 and other laboratory biomarkers and clinical parameters’ section.

7. We note you have included a table to which you do not refer in the text of your manuscript. Please ensure that you refer to Table 2 in your text; if accepted, production will need this reference to link the reader to the Table.

It was a typo and we have corrected it. Table 2 is now referred to in ‘Results- Associations of sST2 with severe adverse events during hospital stay’ section.

8. Please include a copy of Table 3, which you refer to in your text on page 15.

It was a typo and it should be ‘Table 2’. We have corrected it.

A list of supplementary figure and file captions has been added after the reference list. The supplementary figures are cited as S1-S4 Fig in the manuscript. A supporting document containing these figures has been uploaded via the submission system (S1-4_Fig.pdf). Another supporting file (S1_File.pdf) has been added too.

We have revised the manuscript according to the editor and the reviewers’ feedback.

Additional Editor Comments :

1.Severe adverse events were defined as in-hospital death, ICU admission, diagnosis of sepsis or use of extracorporeal membrane oxygenation. Please specify the criteria used to define these, and what about the need for vent support, RRT, etc.

When we designed this study, we conducted a literature search on prognostic studies of biomarkers in both adult and pediatric patients with pneumonia. We found some variations in defining severe short-term outcomes for pneumonia patients. However, most papers used combinations of death, ICU care, sepsis, septic shock, ECMO, mechanical ventilation, vasoactive infusions, chest drainage, respiratory failure, empyema, and ARDS (see, for example, References 4,8,13 and 21 in the manuscript) as outcome factors. For our study, we chose severe adverse events as a composite endpoint, including in-hospital death, ICU admission, diagnosis of sepsis, and use of ECMO—all of which represent death or life-threatening conditions.

1.In-hospital death is death that occurred during hospitalization.

2.In our hospital, a patient is typically admitted to the ICU if at least one of the following conditions is present: (1) FiO2 ≥ 0.6, SaO2 ≤ 0.92, (2) shock and/or impaired consciousness, (3) tachypnea and tachycardia with severe respiratory distress or signs of exhaustions, with or without elevated PaCO2, (4) recurrent apnea, or slow and irregular breathing, and (5) other conditions requiring further monitoring and treatment (e.g. renal failure, severe thrombocytopenia)

3.The definition of sepsis follows the Third International Consensus Definitions for sepsis as life-threatening organ dysfunction caused by a dysregulated host response to infection, where organ dysfunction can be identified as ≥ 2 points on the Phoenix Sepsis Score in children with suspected infection [1].

4.The use of ECMO is mainly indicated for patients requiring respiratory support due to: (1) PaO2/FiO2 < 60-80 mmHg due to severe respiratory failure; (2) failure of conventional ventilation and/or other rescue therapies; (3) high ventilator settings (e.g., mean airway pressure > 20 - 25 cm H₂O during conventional ventilation or > 30 cm H₂O during high-frequency ventilation, or signs of iatrogenic barotrauma). ECMO is also indicated for circulatory support in cases such as cardiogenic shock unresponsive to standard medication, with low SBP < 50 mmHg, urine volume < 1 ml / (kg·h), lactic acidosis, central venous oxygen saturation < 0.6, altered mental status due to low cardiac output, and refractory septic shock with an epinephrine dose > 1 μg / (kg·min) or vasoactive-inotropic score >100.

Regarding renal replacement therapy and ventilation support, we collected information from medical records. Three patients received renal replacement therapy, and these patients were all admitted to the ICU. 6 patients received ventilation support including ECMO (n=4) and invasive mechanical ventilation (n=2). All of these patients were admitted to the ICU. Therefore, all patients who received mechanical ventilation or renal replacement therapy were included in the severe adverse events group.

We reviewed all the patient data to confirm sepsis diagnosis. Based on the diagnostic criteria, nine patients met the definition for sepsis, and all were included in the severe adverse events group.

The manuscript has been revised accordingly. The definitions have been provided in the supporting document ‘S1 File’ cited in the manuscript.

[1] Schlapbach LJ, Watson RS, Sorce LR, Argent AC, Menon K, Hall MW, et al. International Consensus Criteria for Pediatric Sepsis and Septic Shock. JAMA. 2024;331(8):665-674. doi:10.1001/jama.2024.0179

2.In some patients, sepsis might be mild without MODS, others severe sepsis with MODS. Was there any subgroup analysis performed and why if not?

There are nine sepsis patients in our cohort, among whom eight had multiorgan dysfunction. The most common organ dysfunctions happened in respiration system (n = 9) and coagulation system (n = 8), followed by circulation system (n = 4) and renal system (n = 3). One patient had only respiratory failure and received invasive mechanical ventilation. Due to a small sample size of sepsis patients, we did not perform subgroup analysis.

3.You have mentioned that sST2 levels were significantly higher in severe MPP cases and those with co-infections. If co-infections were detected, what is the specificity for its association with MPP? Sst2 could have been elevated due to other infections as well. How do you explain the uncertainty?

In this study, we have demonstrated that sST2 levels were significantly higher in severe MPP, MPP with complications, and MPP with severe adverse events. We consider sST2 to be a prognostic marker, rather than a diagnostic marker. Due to the biological functions of sST2, high sST2 levels may reflect excessive immune response and severe inflammation.

sST2 is not specific to MPP. Previous studies (Reference 12,27) have shown that sST2 levels increase in pneumonia caused by other pathogens. We found that sST2 levels were higher in MPP with co-infections. A higher proportion of patients with co-infections were in the severe MPP group compared to the proportion of patients with MP infection alone, although this difference did not reach statistical significance (37.3% vs. 23.4%, P = 0.0723). We think this may partly explain the association of sST2 and co-infection.

Meanwhile, among the patients with only MP infection, sST2 levels in severe MPP patients were higher than non-severe MPP (30.8 [14.5, 75.4] vs 22.6 [12.8, 52.0] ng/mL, P=0.361), although it did not reach statistical significance, possibly due to a relatively small sample size. Further, among the patients with only MP infection, three patients had severe adverse events and their sST2 levels (49.2, 145.2, and 189.7 ng/mL respectively) were significantly higher than the rest (22.6 [12.8, 52.0] ng/mL), but the sample size is very small. These findings warrant further validation in a larger cohort.

We will conduct a prospective study to evaluate the role of sST2 in children of community acquired pneumonia caused by different pathogens including MP with a larger sample size in the future.

4.You have mentioned that sST2 levels correlated positively with neutrophil count, neutrophil to lymphocyte ratio, PCT, CRP and IL-6- comparative analysis. Was sST2 more specific, more sensitive, cost effective, etc. What is the distinct advantage of doing Sst2, compared to performing NLR, ANC, PCT, CRP?

In this study, we aimed to demonstrate the value of sST2 in predicting severe adverse events as a prognostic marker. Through comparisons of ROC AUCs, we found that sST2 had better performance in predicting severe adverse events in this cohort. Its AUC was significantly higher than those of CRP and NLR, and higher than those of PCT and IL-6 without statistical significance. Furthermore, in our cohort, sST2 was the only independent biomarker associated with severe adverse events in a multiple logistic regression analysis which included sST2, PCT and IL-6 as independent variables (see Results—Associations of sST2 with severe adverse events during hospital stay). Therefore, we think sST2 may predict severe adverse events more effectively than conventional inflammatory biomarkers. However, this retrospective study has several limitations as we have mentioned in our manuscript, therefore we plan to confirm these findings in a prospective study with a larger sample size.

5.Lines 49-50: Studies reported mixing findings on accuracy of this method to predict patients’ outcome and inter-physician variations [4,5]- Grammar error needs to be corrected.

We corrected it as “Previous studies have reported mixed results regarding the accuracy of this approach as well as substantial inter-physician variability”.

6.What is authenticity of definition of severe MPP? Your aim was to evaluate a biomarker whether it predicts severity. If only 4 parameters are assessed, there is a large gap of knowledge about those patients who were oxygen dependent, MODS but did not require organ support, those left with complications, destroyed lungs, etc.

The diagnosis for MPP and severe MPP were established in accordance with the Chinese guideline for the diagnosis and treatment of childhood Mycoplasma pneumoniae pneumonia (2023) (Reference 1, 2). This guideline was issued by the National Health Commission of the People's Republic of China.

The definition of severe MPP is provided in the ‘Materials and Method- Clinical data collection section’, as ‘MPP patients with any of the following criteria[1,2]: (1) high fever (> 39°C) for more than 5 days or fever for more than 7 days without a declining trend in peak temperature; (2) hypoxemia (maintained an SaO2 < 92% on room air); (3) increasing respiratory and pulse rates with clinical evidence of respiratory distress and exhaustion with or without a elevated PaCO2; (4) signs of intrapulmonary infection, such as moderate to large pleural effusion, large area of pulmonary consolidation, plastic bronchitis, pulmonary embolism, necrotizing pneumonia, and acute asthma exacerbations; and (5) signs of extrapulmonary complications, such as meningoencephalitis, ascending (i.e., Guillain-Barré) paralysis, myopericarditis, erythema multiforme, autoimmune hemolytic anemia, hemophagocytic syndrome, or disseminated intravascular coagulation’ According to the above definition, the MPP patients with complications, and destroyed lungs requiring oxygenations are included in severe MPP.

Although our primary aim is to evaluate the association of sST2 with severe adverse events, we evaluated its association with clinical characteristics, including severe MPP. We found that sST2 levels in the severe MPP patients were significantly higher (71.91 [30.76-140.12] vs 26.2 [15.15-54.69], P < 0.001). Among the severe MPP patients, sST2 levels were significantly higher in those with severe adverse events compared to those without (170.7 [132.5, 246.3] vs. 45.1 [18.0, 90.7] ng/mL, P < 0.001).

7.Lines 83-84: The diagnosis of SMPP is made in MPP patients with either of the following criteria [2]- Grammar error needs to be corrected.

We corrected it as “Severe MPP was defined as MPP with any of the following criteria”.

8.Line 140: media hospital stay was 7 days- grammar error

We corrected it as “median hospital stay”.

9.Line 142: ECOM- spelling error needs to be corrected.

We corrected it as “ECMO”.

10.The

---

## [Editor Report · Decision Letter 1]

30 Mar 2026

Dear Dr. YE,

Thank you for submitting your manuscript to PLOS ONE. After careful consideration, we feel that it has merit but does not fully meet PLOS ONE’s publication criteria as it currently stands. Therefore, we invite you to submit a revised version of the manuscript that addresses the points raised during the review process.

As the corresponding author, your ORCID iD is verified in the submission system and will appear in the published article. PLOS supports the use of ORCID, and we encourage all coauthors to register for an ORCID iD and use it as well. Please encourage your coauthors to verify their ORCID iD within the submission system before final acceptance, as unverified ORCID iDs will not appear in the published article. *Only* the individual author can complete the verification step; PLOS staff the individual author can complete the verification step; PLOS staff the individual author can complete the verification step; PLOS staff the individual author can complete the verification step; PLOS staff *cannot* verify ORCID iDs on behalf of authors.verify ORCID iDs on behalf of authors.verify ORCID iDs on behalf of authors.verify ORCID iDs on behalf of authors.

We look forward to receiving your revised manuscript.

Kind regards,

Shivkumar Gopalakrishnan, MD

Academic Editor

PLOS One

Journal Requirements:

Additional Editor Comments:

Dear Dr. YE,

Your article is well conceived and holds promise. However, there are a few uncertainties which demand clarification.

1. Severe adverse events were defined as in-hospital death, ICU admission, diagnosis of sepsis or use of extracorporeal membrane oxygenation. Please specify the criteria used to define these, and what about the need for vent support, RRT, etc.

2. In some patients, sepsis might be mild without MODS, others severe sepsis with MODS. Was there any subgroup analysis performed and why if not?

3. You have mentioned that sST2 levels were significantly higher in severe MPP cases and those with co-infections. If co-infections were detected, what is the specificity for its association with MPP? Sst2 could have been elevated due to other infections as well. How do you explain the uncertainty?

4. You have mentioned that sST2 levels correlated positively with neutrophil count, neutrophil to lymphocyte ratio, PCT, CRP and IL-6- comparative analysis. Was sST2 more specific, more sensitive, cost effective, etc. What is the distinct advantage of doing Sst2, compared to performing NLR, ANC, PCT, CRP?

5. Lines 49-50: Studies reported mixing findings on accuracy of this method to predict patients’ outcome and inter-physician variations [4,5]- Grammar error needs to be corrected.

6. What is authenticity of definition of severe MPP? Your aim was to evaluate a biomarker whether it predicts severity. If only 4 parameters are assessed, there is a large gap of knowledge about those patients who were oxygen dependent, MODS but did not require organ support, those left with complications, destroyed lungs, etc.

7. Lines 83-84: The diagnosis of SMPP is made in MPP patients with either of the following criteria [2]- Grammar error needs to be corrected.

8. Line 140: media hospital stay was 7 days- grammar error

9. Line 142: ECOM- spelling error needs to be corrected.

10. The study design is ambiguous- criteria for severe MPP is delineated, however, the parameter Sst2 was evaluated only for 12 patients who had fit into 4 separate set of “adverse events”. These adverse events are overlapping over the defined criteria. It is unclear about sst2 levels among the actual population of severe MPP patients who qualified based on set criteria [2]. The study is actually assessing predictive capacity of sst2 to identify severe adverse events among MPP rather than severity of MPP.

11. Line 229: Consequently, MP infection undergone a prolonged non-seasonal epidemic, which emerged in April 2023- grammar error needs correction.

12. Title: Soluble ST2 is a potential biomarker for risk stratification of pediatric patients with mycoplasma pneumoniae pneumonia- better stated as sst2 as a biomarker for predicting severe adverse events among paediatric patients with MPP.

Thank you, with regards, Dr SHIVKUMAR GOPALAKRISHNAN, MD.,

---

## [Author Response · Author response to Decision Letter 2]

1 Apr 2026

PONE-D-25-53987

Soluble ST2 is a potential biomarker for risk stratification of pediatric patients with mycoplasma pneumoniae pneumonia

PLOS One

Dear Editor and Reviewers,

Thank you very much for your valuable comments and constructive suggestions on our manuscript. We have carefully gone over all the feedback and revised the manuscript accordingly. The replies to all the comments are provided in the uploaded response document. A revised manuscript has been uploaded too.

We appreciate all your time and efforts in helping us make this study and this manuscript better.

Best regards,

Zhicheng Ye

The responses to the editor and the reviewers are as the below in blue.

Journal Requirements:

We have reformatted the manuscript according to the two style templates.

The ORCID iDs for the corresponding authors have been updated via submission system

（https://orcid.org/my-orcid?orcid=0009-0009-6967-6969，ORCID：0009-0009-6967-6969）.

This study was approved by the Ethics Committee of the Children’s Hospital of Fudan University (Ethics No:2024-129), and need for consent was waived by the ethics committee. We have included an Ethics Statement subsection in the Materials and methods: “This study was conducted in accordance with the Declaration of Helsinki and was approved by the Ethics Committee of the Children’s Hospital of Fudan University (Ethics No. 2024-129). We confirm that all methods were performed in accordance with the relevant guidelines and regulations. This study is a retrospective analysis, only involving the statistical analysis of patient data, and all patient information has been anonymized and does not contain sensitive data; therefore, the requirement for informed consent was waived by the ethics committee.”

The same text has been added to the “Ethics Statement” field of the submission form via the submission system.

As described above, the full ethics statement has been provided in the ‘Materials and methods-Ethics Statement’ and via the online submission system.

5. We notice that your supplementary [figures] are included in the manuscript file. Please remove them and upload them with the file type 'Supporting Information'. Please ensure that each Supporting Information file has a legend listed in the manuscript after the references list.

The supplementary figures have been removed from the manuscript file and a list of supplementary figure captions has been added in the manuscript after the reference list. A supporting document named S1-4_Fig.pdf is uploaded via the submission system.

6. Please ensure that you refer to Figure 2 in your text as, if accepted, production will need this reference to link the reader to the figure.

It was a typo and we have corrected it. Figure 2 is now referred to in ‘Results- Correlation between sST2 and other laboratory biomarkers and clinical parameters’ section.

7. We note you have included a table to which you do not refer in the text of your manuscript. Please ensure that you refer to Table 2 in your text; if accepted, production will need this reference to link the reader to the Table.

It was a typo and we have corrected it. Table 2 is now referred to in ‘Results- Associations of sST2 with severe adverse events during hospital stay’ section.

8. Please include a copy of Table 3, which you refer to in your text on page 15.

It was a typo and it should be ‘Table 2’. We have corrected it.

A list of supplementary figure and file captions has been added after the reference list. The supplementary figures are cited as S1-S4 Fig in the manuscript. A supporting document containing these figures has been uploaded via the submission system (S1-4_Fig.pdf). Another supporting file (S1_File.pdf) has been added too.

We have revised the manuscript according to the editor and the reviewers’ feedback.

Additional Editor Comments :

1.Severe adverse events were defined as in-hospital death, ICU admission, diagnosis of sepsis or use of extracorporeal membrane oxygenation. Please specify the criteria used to define these, and what about the need for vent support, RRT, etc.

When we designed this study, we conducted a literature search on prognostic studies of biomarkers in both adult and pediatric patients with pneumonia. We found some variations in defining severe short-term outcomes for pneumonia patients. However, most papers used combinations of death, ICU care, sepsis, septic shock, ECMO, mechanical ventilation, vasoactive infusions, chest drainage, respiratory failure, empyema, and ARDS (see, for example, References 4,8,13 and 21 in the manuscript) as outcome factors. For our study, we chose severe adverse events as a composite endpoint, including in-hospital death, ICU admission, diagnosis of sepsis, and use of ECMO—all of which represent death or life-threatening conditions.

1.In-hospital death is death that occurred during hospitalization.

2.In our hospital, a patient is typically admitted to the ICU if at least one of the following conditions is present: (1) FiO2 ≥ 0.6, SaO2 ≤ 0.92, (2) shock and/or impaired consciousness, (3) tachypnea and tachycardia with severe respiratory distress or signs of exhaustions, with or without elevated PaCO2, (4) recurrent apnea, or slow and irregular breathing, and (5) other conditions requiring further monitoring and treatment (e.g. renal failure, severe thrombocytopenia)

3.The definition of sepsis follows the Third International Consensus Definitions for sepsis as life-threatening organ dysfunction caused by a dysregulated host response to infection, where organ dysfunction can be identified as ≥ 2 points on the Phoenix Sepsis Score in children with suspected infection [1].

4.The use of ECMO is mainly indicated for patients requiring respiratory support due to: (1) PaO2/FiO2 < 60-80 mmHg due to severe respiratory failure; (2) failure of conventional ventilation and/or other rescue therapies; (3) high ventilator settings (e.g., mean airway pressure > 20 - 25 cm H₂O during conventional ventilation or > 30 cm H₂O during high-frequency ventilation, or signs of iatrogenic barotrauma). ECMO is also indicated for circulatory support in cases such as cardiogenic shock unresponsive to standard medication, with low SBP < 50 mmHg, urine volume < 1 ml / (kg·h), lactic acidosis, central venous oxygen saturation < 0.6, altered mental status due to low cardiac output, and refractory septic shock with an epinephrine dose > 1 μg / (kg·min) or vasoactive-inotropic score >100.

Regarding renal replacement therapy and ventilation support, we collected information from medical records. Three patients received renal replacement therapy, and these patients were all admitted to the ICU. 6 patients received ventilation support including ECMO (n=4) and invasive mechanical ventilation (n=2). All of these patients were admitted to the ICU. Therefore, all patients who received mechanical ventilation or renal replacement therapy were included in the severe adverse events group.

We reviewed all the patient data to confirm sepsis diagnosis. Based on the diagnostic criteria, nine patients met the definition for sepsis, and all were included in the severe adverse events group.

The manuscript has been revised accordingly. The definitions have been provided in the supporting document ‘S1 File’ cited in the manuscript.

[1] Schlapbach LJ, Watson RS, Sorce LR, Argent AC, Menon K, Hall MW, et al. International Consensus Criteria for Pediatric Sepsis and Septic Shock. JAMA. 2024;331(8):665-674. doi:10.1001/jama.2024.0179

2.In some patients, sepsis might be mild without MODS, others severe sepsis with MODS. Was there any subgroup analysis performed and why if not?

There are nine sepsis patients in our cohort, among whom eight had multiorgan dysfunction. The most common organ dysfunctions happened in respiration system (n = 9) and coagulation system (n = 8), followed by circulation system (n = 4) and renal system (n = 3). One patient had only respiratory failure and received invasive mechanical ventilation. Due to a small sample size of sepsis patients, we did not perform subgroup analysis.

3.You have mentioned that sST2 levels were significantly higher in severe MPP cases and those with co-infections. If co-infections were detected, what is the specificity for its association with MPP? Sst2 could have been elevated due to other infections as well. How do you explain the uncertainty?

In this study, we have demonstrated that sST2 levels were significantly higher in severe MPP, MPP with complications, and MPP with severe adverse events. We consider sST2 to be a prognostic marker, rather than a diagnostic marker. Due to the biological functions of sST2, high sST2 levels may reflect excessive immune response and severe inflammation.

sST2 is not specific to MPP. Previous studies (Reference 12,27) have shown that sST2 levels increase in pneumonia caused by other pathogens. We found that sST2 levels were higher in MPP with co-infections. A higher proportion of patients with co-infections were in the severe MPP group compared to the proportion of patients with MP infection alone, although this difference did not reach statistical significance (37.3% vs. 23.4%, P = 0.0723). We think this may partly explain the association of sST2 and co-infection.

Meanwhile, among the patients with only MP infection, sST2 levels in severe MPP patients were higher than non-severe MPP (30.8 [14.5, 75.4] vs 22.6 [12.8, 52.0] ng/mL, P=0.361), although it did not reach statistical significance, possibly due to a relatively small sample size. Further, among the patients with only MP infection, three patients had severe adverse events and their sST2 levels (49.2, 145.2, and 189.7 ng/mL respectively) were significantly higher than the rest (22.6 [12.8, 52.0] ng/mL), but the sample size is very small. These findings warrant further validation in a larger cohort.

We will conduct a prospective study to evaluate the role of sST2 in children of community acquired pneumonia caused by different pathogens including MP with a larger sample size in the future.

4.You have mentioned that sST2 levels correlated positively with neutrophil count, neutrophil to lymphocyte ratio, PCT, CRP and IL-6- comparative analysis. Was sST2 more specific, more sensitive, cost effective, etc. What is the distinct advantage of doing Sst2, compared to performing NLR, ANC, PCT, CRP?

In this study, we aimed to demonstrate the value of sST2 in predicting severe adverse events as a prognostic marker. Through comparisons of ROC AUCs, we found that sST2 had better performance in predicting severe adverse events in this cohort. Its AUC was significantly higher than those of CRP and NLR, and higher than those of PCT and IL-6 without statistical significance. Furthermore, in our cohort, sST2 was the only independent biomarker associated with severe adverse events in a multiple logistic regression analysis which included sST2, PCT and IL-6 as independent variables (see Results—Associations of sST2 with severe adverse events during hospital stay). Therefore, we think sST2 may predict severe adverse events more effectively than conventional inflammatory biomarkers. However, this retrospective study has several limitations as we have mentioned in our manuscript, therefore we plan to confirm these findings in a prospective study with a larger sample size.

5.Lines 49-50: Studies reported mixing findings on accuracy of this method to predict patients’ outcome and inter-physician variations [4,5]- Grammar error needs to be corrected.

We corrected it as “Previous studies have reported mixed results regarding the accuracy of this approach as well as substantial inter-physician variability”.

6.What is authenticity of definition of severe MPP? Your aim was to evaluate a biomarker whether it predicts severity. If only 4 parameters are assessed, there is a large gap of knowledge about those patients who were oxygen dependent, MODS but did not require organ support, those left with complications, destroyed lungs, etc.

The diagnosis for MPP and severe MPP were established in accordance with the Chinese guideline for the diagnosis and treatment of childhood Mycoplasma pneumoniae pneumonia (2023) (Reference 1, 2). This guideline was issued by the National Health Commission of the People's Republic of China.

The definition of severe MPP is provided in the ‘Materials and Method- Clinical data collection section’, as ‘MPP patients with any of the following criteria[1,2]: (1) high fever (> 39°C) for more than 5 days or fever for more than 7 days without a declining trend in peak temperature; (2) hypoxemia (maintained an SaO2 < 92% on room air); (3) increasing respiratory and pulse rates with clinical evidence of respiratory distress and exhaustion with or without a elevated PaCO2; (4) signs of intrapulmonary infection, such as moderate to large pleural effusion, large area of pulmonary consolidation, plastic bronchitis, pulmonary embolism, necrotizing pneumonia, and acute asthma exacerbations; and (5) signs of extrapulmonary complications, such as meningoencephalitis, ascending (i.e., Guillain-Barré) paralysis, myopericarditis, erythema multiforme, autoimmune hemolytic anemia, hemophagocytic syndrome, or disseminated intravascular coagulation’ According to the above definition, the MPP patients with complications, and destroyed lungs requiring oxygenations are included in severe MPP.

Although our primary aim is to evaluate the association of sST2 with severe adverse even

---

## [Editor Report · Decision Letter 2]

7 Apr 2026

Soluble ST2 as a biomarker for predicting severe adverse events among pediatric patients with Mycoplasma pneumoniae pneumonia

PONE-D-25-53987R2

Dear Dr. ZHICHENG YE,

We’re pleased to inform you that your manuscript has been judged scientifically suitable for publication and will be formally accepted for publication once it meets all outstanding technical requirements.

Kind regards,

Shivkumar Gopalakrishnan, MD

Academic Editor

PLOS One
---

## [Editor Report · Acceptance letter]

PONE-D-25-53987R2

PLOS One

Dear Dr. YE,

I'm pleased to inform you that your manuscript has been deemed suitable for publication in PLOS One. Congratulations! Your manuscript is now being handed over to our production team.

Kind regards,

on behalf of

Dr. Shivkumar Gopalakrishnan

Academic Editor

PLOS One